# DIFFUSION MODELS NEED VISUAL PRIORS FOR IMAGE GENERATION

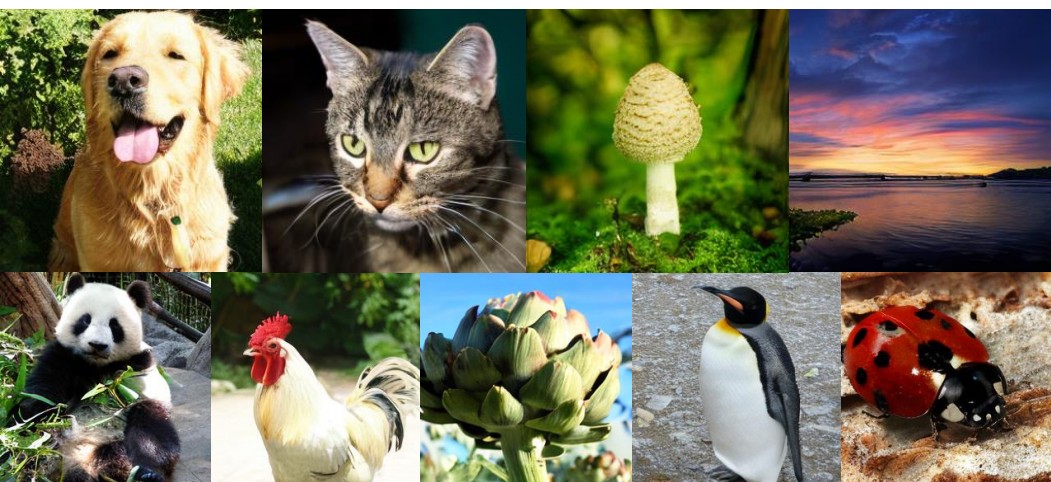

Figure 1: **Selected samples generated by the second stage of DoD-XL.** By training for only 1 million steps on ImageNet-256 × 256 dataset, DoD-XL achieves state-of-the-art image quality.

## ABSTRACT

Conventional class-guided diffusion models generally succeed in generating images with correct semantic content, but often struggle with texture details. This limitation stems from the usage of class priors, which only provide coarse and limited conditional information. To address this issue, we propose Diffusion on Diffusion (DoD), an innovative multi-stage generation framework that first extracts visual priors from previously generated samples, then provides rich guidance for the diffusion model leveraging visual priors from the early stages of diffusion sampling. Specifically, we introduce a latent embedding module that employs a compression-reconstruction approach to discard redundant detail information from the conditional samples in each stage, retaining only the semantic information for guidance. We evaluate DoD on the popular ImageNet-256 × 256 dataset, reducing 7× training cost compared to SiT and DiT with even better performance in terms of the FID-50K score. Our largest model DoD-XL achieves an FID-50K score of 1.83 with only 1 million training steps, which surpasses other state-of-the-art methods without bells and whistles during inference.

## 1 INTRODUCTION

Diffusion models have emerged as a paradigm-shifting approach in visual content generation employing an innovative process of iterative noise-to-data transformation. Trained to reverse a gradual noising process, these models leverage deep neural networks to generate high-quality new samples that faithfully represent the training data distribution. Diffusion models have surpassed previous state-of-the-art generative frameworks, such as GANs (Sauer et al., 2022; Goodfellow et al., 2014; 2020) and VAEs (Kingma & Welling, 2013), offering superior sample quality, improved training stability, and enhanced scalability. This superiority of diffusion models has led to their widespread

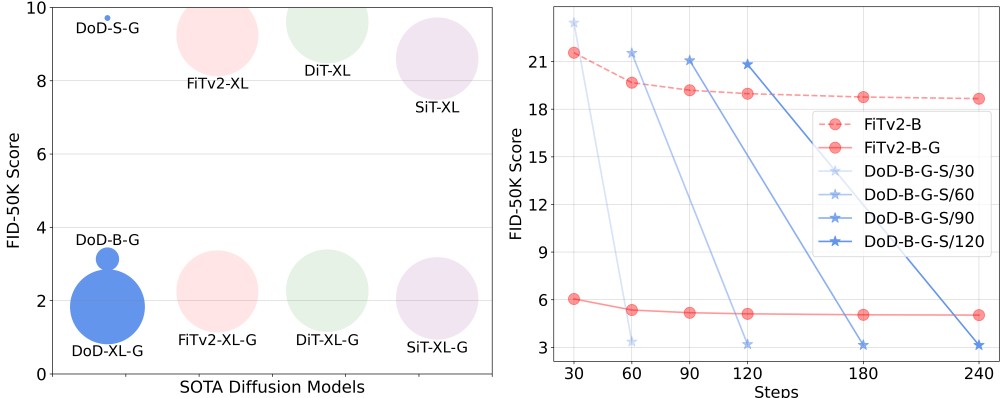

Figure 2: **ImageNet generation with Diffusion on Diffusion (DoD).** *Left:* **DoD is parameter-efficient.** The diameter of each circle indicates the model size. Our DoD-S and DoD-B models, despite being much smaller, are comparable to the XL variants of other diffusion models. *Right:* **DoD is sampling-efficient.** "-G" indicates the application of classifier-free guidance (CFG), and "-S" denotes the number of sampling steps for each stage of DoD. The starting and ending points of each blue line represent the two stages of DoD, where we only apply CFG in the second stage. With the same sampling steps, DoD achieves lower FID-50K score.

adoption across a diverse range of conditional generation tasks, including class-guided image generation (Lu et al., 2024a; Wang et al., 2024; Ma et al., 2024; Peebles & Xie, 2023), text-to-image generation (Esser et al., 2024b; Podell et al., 2023), and image editing (Meng et al., 2021).

Conventionally, class-guided diffusion models generate images conditioned on learned class embeddings. While this embedding-based approach is widely adopted, such class prior can only provide coarse-grained conditional information for models, which is only able to distinguish different categories. The challenge of constructing detailed images from such limited priors has led to exceptionally long training cycles for current class-guided diffusion models. For example, DiT (Peebles & Xie, 2023) and SiT (Ma et al., 2024) require up to 7 million training steps to achieve convergence.

In contrast, visual priors contain more geometric visual information. Intuitively, visual priors should be closer to the target distribution of image generation. The efficacy of visual priors in enhancing image generation quality has been demonstrated in various domains, including super-resolution models (Yang et al., 2024; Ren et al., 2024) and SD-Edit (Meng et al., 2021). Inspired by this well-established methodology, our study seeks to explore *the integration of visual priors into class-guided image generation models.* To this end, we propose an innovative multi-stage diffusion sampling framework. The initial stage adheres to the conventional approach, utilizing fixed class embeddings as priors. However, the subsequent refinement stages employ the image generated in the previous stage as a visual prior to guide further image synthesis. Our framework's distinctive feature lies in the reuse of the same diffusion model across multiple stages, leading us to term this method "Diffusion on Diffusion" (DoD).

The proposed Diffusion on Diffusion (DoD) framework introduces a recurrent approach, where each stage comprises a complete diffusion sampling procedure. From the second stage, DoD extracts semantic information from the previous output as additional visual priors. This mechanism provides rich semantic visual guidance during the early stages of diffusion sampling, facilitating the generation of higher-quality images. By repeatedly leveraging the generation capabilities of the diffusion model, DoD not only enhances texture details but also refines the object-level geometric. Although longer sampling steps in diffusion models typically provide more accurate approximations and potentially boost performance, the final performance is still constrained by the model capacity. In contrast, our method, by incorporating visual priors in extended sampling steps, effectively improves both the sampling efficiency and generation quality. As shown in Figure 2 (*Right*), the proposed paradigm yields more efficient sampling compared to simply increasing the sample steps in diffusion models. Moreover, as illustrated in Figure 2 (*Left*), DoD is also parameter-efficient. Unlike models such as SDXL (Podell et al., 2023) which employs a separate refiner model, DoD integrates both image generation and refinement using shared parameters, significantly reducing the model size and lowering the barrier to real-world applications.

Directly using the generated latents as conditions results in the model collapsing into a complete reconstruction of the input samples, thereby losing the desired refinement effect. To overcome this issue, we introduce a Latent Embedding Module (LEM), a vision transformer, that compresses the conditional sample into a few low-dimensional vectors, thereby discarding redundant texture details while retaining the essential information. Our investigation shows that the kept information primarily consists of semantics, which are aligned between the training dataset and the generated samples for a well-trained generation model. The semantic alignment eliminates the need for collecting specialized refinement data or implementing complex training strategies, allowing DoD to be trained end-to-end on generation datasets.

DoD is built upon the state-of-the-art diffusion transformer, FiTv2 (Wang et al., 2024), and follows the latent diffusion model (LDM) (Rombach et al., 2022) training paradigm. We conduct comprehensive experiments and strictly evaluate our proposed method on *ImageNet*-$256 \times 256$ benchmark. Compared with the DiT (Peebles & Xie, 2023) and SiT (Ma et al., 2024) models, our DoD achieves even better performance in terms of FID, with fewer model parameters and computational complexity, consuming $7\times$ less training cost. Meanwhile, our method outperforms the previous state-of-the-art methods when no bells and whistles were applied during inference, which achieves an FID-50K score of 1.83 with only 1 million training steps.

## 2 RELATED WORK

### 2.1 DIFFUSIONS AND FLOWS

Denoising Diffusion Probabilistic Models (DDPMs) (Ho et al., 2020; Saharia et al., 2022; Radford et al., 2021; Croitoru et al., 2023; Bond-Taylor et al., 2021) and score-based models (Hyvärinen & Dayan, 2005; Song et al., 2020b) have demonstrated significant advancements in image generation tasks (Lu et al., 2024b; Ling et al., 2024; Rombach et al., 2022; Saharia et al., 2022; Meng et al., 2021; Ramesh et al., 2022; Ruiz et al., 2023; Poole et al., 2022). The Denoising Diffusion Implicit Model (DDIM) (Song et al., 2020a) introduced an accelerated sampling method, while Latent Diffusion Models (LDMs) (Rombach et al., 2022) set a new standard by applying deep generative models to reverse the noise process in the latent space using Variational Autoencoders (VAEs) (Kingma & Welling, 2013). Flow models (Liu et al., 2023; Albergo & Vanden-Eijnden, 2022; Lipman et al., 2022; Albergo et al., 2023) present an alternative approach by learning a neural ordinary differential equation (ODE) that transports between two distributions. The rectified flow model (Liu et al., 2023) solves a nonlinear least squares optimization problem to learn mappings along straight paths, which represent the shortest distance between two points, leading to improved computational efficiency. In this work, we adopt the rectified flow as the noise scheduler to train our DoD models.

### 2.2 DIFFUSION TRANSFORMER

The Transformer model (Vaswani et al., 2017) has successfully replaced domain-specific architectures across various fields, including language (Brown et al., 2020; Chowdhery et al., 2023), vision (Dosovitskiy et al., 2020; Han et al., 2022), and multi-modal learning (Team et al., 2023). In the realm of visual perception research, numerous studies (Touvron et al., 2019; 2021; Liu et al., 2021; 2022) have focused on accelerating pretraining by utilizing fixed, low-resolution images. Transformers have also been applied in denoising diffusion probabilistic models (Ho et al., 2020) for image synthesis. DiT (Peebles & Xie, 2023), a pioneering work in this space, employs a vision transformer as the backbone for latent diffusion models (LDMs), serving as a strong baseline for subsequent research. MDT (Gao et al., 2023) introduces a masked latent modeling approach, requiring two forward passes during training and inference. U-ViT (Bao et al., 2023) tokenizes all inputs and integrates U-Net architectures into the ViT backbone of LDMs. SiT (Ma et al., 2024), utilizing the same architecture as DiT, explores various rectified flow configurations. Large-DiT and Flag-DiT (Gao et al., 2024) scale up diffusion transformers to achieve improved performance. SD3 (Esser et al., 2024a) introduces novel noise samplers for rectified flow models and scales these models to billions of parameters, yielding state-of-the-art text-to-image generation results. FiTv2 (Wang et al., 2024), based on FiT (Lu et al., 2024a), achieves advanced class-conditional image generation performance by leveraging a flexible diffusion transformer architecture within a rectified flow framework. In this work, built upon the FiTv2 architecture, we propose Diffusion on Diffusion (DoD), an innovative framework which effectively incorporates visual priors into class-guided image generation.

## 3 METHOD

### 3.1 PRELIMINARIES

**Diffusion and Flow Models.** Before introducing our Diffusion on Diffusion (DoD) framework, we provide a brief review of diffusion and flow models. Given the noise distribution $\epsilon \sim \mathcal{N}(0, \mathbf{I})$ and data distribution $x_0 \sim p(x)$, the models use the time-dependent forward process:

$$x_t = \alpha_t x_0 + \beta_t \epsilon, \tag{1}$$

where $\alpha_t$ is a decreasing function of $t$ and $\beta_t$ is an increasing function of $t$. In this unified perspective, diffusion models (Ho et al., 2020; Song et al., 2020b; Song & Ermon, 2019; 2020) set $\alpha_t$ and $\beta_t$ based on stochastic differential equation (SDE) formulations, where DDPM (Ho et al., 2020) is equivalent to variance preserving SDE (VP-SDE) and SMLD (Song & Ermon, 2019; 2020) corresponds to variance exploding SDE (VE-SDE). DDIM (Song et al., 2020a) sets $\alpha_t$ and $\beta_t$ through ordinary differential equations (ODE), which leads to fewer sampling steps but sacrifices the generation quality. Flow models (Liu et al., 2023; Lipman et al., 2022; Albergo & Vanden-Eijnden, 2022; Albergo et al., 2023) restrict the process 1 on $t \in [0, 1]$, and set $\alpha_0 = \beta_1 = 1, \alpha_1 = \beta_0 = 0$, interpolating between the two distributions through ODE formulation.

Rectified flow (Liu et al., 2023) introduces an ODE model and transports between the data distribution and the noise distribution via a straight line path, which is the theoretically shortest route between two points. Given empirical observations $X_0 \sim p(x), X_1 \sim \mathcal{N}(0, \mathbf{I})$, the forward process 1 is defined as: $X_t := tX_1 + (1-t)X_0$, which is the linear interpolation of $X_0$ and $X_1$.

The ODE model learns the drift at time $t \in [0, 1]$:

$$\mathrm{d}X_t = v(X_t, t)\mathrm{d}t, \tag{2}$$

which converts data $X_0$ from $p(x)$ to noise $X_1$ from $\pi_1$, and the drift follows the direction of $(X_1 - X_0)$. In practice, a network $v_\theta$ is utilized to predict this drift, and the optimization target is:

$$\min_v \int_0^1 \mathbb{E}\Big[||(X_1 - X_0) - v(X_t, t)||^2\Big]\mathrm{d}t \tag{3}$$

Previous studies (Ma et al., 2024; Esser et al., 2024b; Wang et al., 2024; Gao et al., 2024) have demonstrated the efficiency and stability of rectified flow models. In this work, we fully adhere to the form of rectified flow as our noise scheduler and enhance the diffusion model by adding extra conditions beyond the class label. We adopt the implementation of rectified flow following SiT (Ma et al., 2024) and use ODE sampler for image synthesis.

**FiTv2.** DoD utilizes the state-of-the-art diffusion transformer, FiTv2 (Wang et al., 2024), as the backbone. FiTv2 is an advanced diffusion transformer on class-guided image generation, evolving from SiT (Ma et al., 2024) and FiT (Lu et al., 2024a). The key modules of FiTv2 include 2-D Rotary Positional Embedding (2-D RoPE) (Su et al., 2024), Swish-Gated Linear Unit (SwiGLU) (Shazeer, 2020), Query-Key Vector Normalization (QK-Norm), and Adaptive Layer Normalization with Low-Rank Adaptation (AdaLN-LoRA) (Hu et al., 2022).

Despite these advanced modules, FiTv2 adopts the Logit-Normal sampling (Esser et al., 2024b) strategy to accelerate the model convergence. This sampling strategy puts more attention on the middle part of the sampling process, as recent studies (Karras et al., 2022; Chen, 2023) have disclosed that the intermediate part is the most challenging part of the diffusion process.

### 3.2 DIFFUSION ON DIFFUSION WITH VISUAL PRIORS

We introduce the Diffusion on Diffusion (DoD) framework, which enhances diffusion models by recurrently incorporating previously generated samples as visual priors to guide the subsequent sampling process. There are two core components in DoD: the Multi-Stage Sampling strategy and the Latent Embedding Module (LEM). The multi-stage sampling enables the use of visual priors, while the LEM extracts semantic information from the samples generated in the previous stage.

**Multi-Stage Sampling.** The sampling process of DoD consists of multiple stages, with each stage being a complete diffusion sampling process conditioned on different information. As depicted in

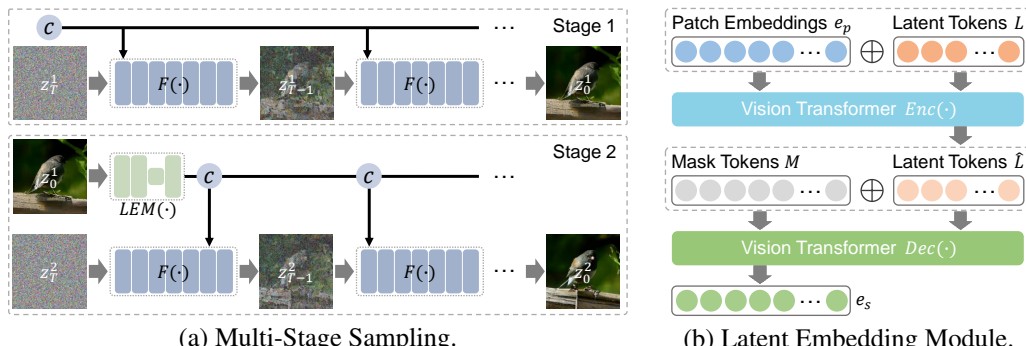

(a) Multi-Stage Sampling.  (b) Latent Embedding Module.

Figure 3: **Illustration of (a) Multi-Stage Sampling and (b) Latent Embedding Module.** We only present the first two stages of DoD, while subsequent stages are derived from the second one.

Figure 3 (a), in the $i$-th stage, the backbone diffusion model, denoted as $F(\cdot)$, takes Gaussian noise $z_1^i \in \mathbb{R}^{H \times W \times d_z}$ as input and progressively denoises it to obtain the sample $z_0^i \in \mathbb{R}^{H \times W \times d_z}$, where $H$, $W$, and $d_z$ denote the height, width, and channels of the noised data, respectively. Let $c$ represent the conditional feature that guides the generation direction via the AdaLN-LoRA block, we have the sampling formulation as:

$$z_{\tau_{t-1}}^i = F(z_{\tau_t}^i, c), t = T, T-1, \ldots, 1, \tag{4}$$

where $T$ represents the total sampling steps, $\tau_t$ is a monotone increasing function of $t$ that satisfies $\tau_t \in [0,1], \tau_0 = 0, \tau_T = 1$. The accurate form of $\{\tau_t\}_{1:T}$ is determined by the ODE sampler. The formulation of each stage closely follows that of FiTv2 (Wang et al., 2024).

Typically, for class-conditional generation, the conditional feature $c$ consists only of class and time information. In DoD, the sample generated in the previous iteration is used as additional conditioning information, as follows:

$$c = \begin{cases} e_c + e_t + \mathbf{S} & \text{if } i = 1 \\ e_c + e_t + e_s^{i-1} & \text{if } i > 1 \end{cases}, \tag{5}$$
$$e_s^{i-1} = \text{LEM}(z_0^{i-1}),$$

where $e_c \in \mathbb{R}^{1 \times d}$ and $e_t \in \mathbb{R}^{1 \times d}$ are the embeddings for the label and time, and $e_s^{i-1} \in \mathbb{R}^{(\frac{H}{p} \times \frac{W}{p}) \times d}$ represents the embedding extracted by the latent embedding module $\text{LEM}(\cdot)$, with $p$ and $d$ denoting the patch size and hidden size of the backbone, respectively.

To maintain consistency in behavior, in the first stage, we replace $e_s$ with a trainable sample token $\mathbf{S} \in \mathbb{R}^{1 \times d}$. Note that when $i > 1$, broadcast is operated on $e_c$ and $e_t$ to align their dimensions with the sample embedding $e_s$. We modify the modulation function in the AdaLN-LoRA block to accept conditional features with different dimensions accordingly.

**Latent Embedding Module (LEM).** Since we use previously generated latents as visual priors to guide the generation process, retaining all the information from these samples would cause the diffusion model to collapse into an identity mapping, resulting in a complete reconstruction of the conditional samples. To avoid this, we propose the Latent Embedding Module (LEM) that filters the conditional information using a compression-reconstruction approach to discard redundant details.

The architecture of LEM is illustrated in Figure 3 (b), comprising an encoder and a decoder, both of which are standard vision transformers (Dosovitskiy et al., 2020). The encoder first patchifies the input latent $z_0 \in \mathbb{R}^{H \times W \times d_z}$ through a patch embedding layer into $e_p \in \mathbb{R}^{(\frac{H}{p} \times \frac{W}{p}) \times d_l}$, where $d_l$ is the hidden size of the LEM, and the patch size $p$ is consistent with that of the backbone network. Next, $N$ learnable latent tokens $\mathbf{L} \in \mathbb{R}^{N \times d_l}$ are concatenated with $e_p$ and fed into the vision transformer. A linear layer then reduces the channel dimension to $d_e$. Only the latent tokens are used as the output of the encoder, formally:

$$\hat{\mathbf{L}} = Enc((e_p + \mathbf{PE}) \oplus \mathbf{L}), \tag{6}$$

Table 1: **Details of DoD models.** The layers of the latent embedding module are presented as encoder depth + decoder depth. We list the number of parameters for the backbone and the latent embedding module in different colors.

| Model | Backbone | | | Latent Embedding Module | | | Params (M) |
|---|---|---|---|---|---|---|---|
| | Layers | Hidden size | Heads | Layers | Hidden size | Heads | |
| DoD-S | 12 | 384 | 6 | 8 + 4 | 384 | 6 | 27 + 21 = 48 |
| DoD-B | 12 | 768 | 12 | 8 + 4 | 768 | 12 | 105 + 86 = 191 |
| DoD-XL | 28 | 1152 | 16 | 8 + 4 | 768 | 12 | 527 + 86 = 613 |

where $\oplus$ denotes concatenation, $\mathbf{PE}$ is the frozen sine-cosine positional embedding, and $Enc(\cdot)$ indicates the encoder. $\hat{\mathbf{L}} \in \mathbb{R}^{N \times d_e}$ represents the output latent tokens, providing a compressed representation of the input latent $z_0$.

The decoder seeks to map the compressed latent tokens $\hat{\mathbf{L}}$ back to the shape of the patch embedding $e_p$ to enable element-wise conditioning. Specifically, a sequence of mask tokens $\mathbf{M} \in \mathbb{R}^{(\frac{H}{p} \times \frac{W}{p}) \times d_e}$, obtained by repeating a shared mask token, is concatenated with $\hat{\mathbf{L}}$. The combined representation is then projected back to $d_l$ and fed into the vision transformer:

$$e_s = Dec((\mathbf{M} + \mathbf{PE}) \oplus \hat{\mathbf{L}}), \tag{7}$$

where $Dec(\cdot)$ denotes the decoder, which has the same number of heads and hidden size as the encoder. The output $e_s$ is the extracted latent condition.

Equipped with the Latent Embedding Module (LEM), DoD realizes image refinement effectively through latent space reconstruction. The encoder of LEM projects the input latents into a more compact feature space, which is then restored by the decoder and the diffusion model. We empirically find that the compressed latent tokens $\hat{\mathbf{L}}$ primarily retain semantic information, enabling the diffusion model to focus on texture details, thereby enhancing image fidelity, as in section 4.2.

### 3.3 CONFIGURATIONS

**Training.** DoD is trained within the latent space of a pre-trained Variational Autoencoder (VAE) from Stable Diffusion (Rombach et al., 2022). The VAE encoder maps RGB images of shape $256 \times 256 \times 3$ to latents of shape $32 \times 32 \times 4$. The new latents sampled by DoD are then decoded back to pixel space by the VAE decoder.

As mentioned above, the Latent Embedding Module (LEM) extracts semantic information from samples through compression and reconstruction to serve as visual priors. We reasonably assume that the high-level semantic information extracted from generated images is similar to that obtained from real images. This assumption allows us to use the latents of ground truth images as inputs to LDM during training, simplifying the training strategy. Such simplification allows end-to-end training of DoD on image latents and joint optimization of the backbone model and LEM. Similar to training with classifier-free guidance, we randomly replace the output of the latent embedding module with a trainable sample token $\mathbf{S}$ at a probability of $p_s$, which is set to 0.5 by default.

**Sampling.** Classifier-free guidance (CFG) (Ho & Salimans, 2021) is well-known for enhancing generation quality and improving the alignment between conditions and generated images. For visual priors, we employ a large CFG scale to ensure the consistency of the generated samples, while we do not use CFG in the first stage of DoD. Additionally, DoD employs the adaptive-step ODE sampler (*i.e.*, dopri5) same as SiT (Ma et al., 2024) for sample synthesis in each stage.

**Models.** We use three different model configurations with varying sizes, DoD-{S, B, XL}, as detailed in Table 1. For the backbone network, *i.e.*, FiTv2, we closely follow the configurations outlined in their paper but reduce the model depth to match those of DiT (Peebles & Xie, 2023) and SiT (Ma et al., 2024). For the latent embedding module, we use an 8-layer encoder and a 4-layer decoder by default. Our largest model, DoD-XL, uses the same size latent embedding module as DoD-B to reduce the number of parameters. The patch size for both the backbone network and the latent embedding module is set to 2, and the channel dimension $d_e$ of the latent tokens in the latent embedding module is set to 16.

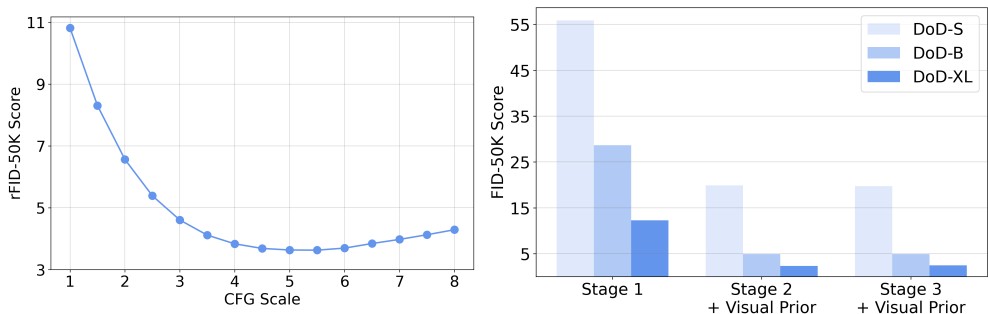

Figure 4: *Left:* **rFID score with different CFG scales.** A large CFG scale is required in DoD to effectively leverage visual priors. *Right:* **FID score comparison across different stages.** Visual priors are not available in Stage 1, while both stages 2 & 3 utilize the samples from the previous stage as visual priors.

## 4 EXPERIMENTS

### 4.1 EXPERIMENTAL SETUP

**Implementation Details.** We conduct experiments on ImageNet (Deng et al., 2009), using a resolution of $256 \times 256$. All models share the same training strategy. We use the AdamW (Kingma, 2014; Loshchilov & Hutter, 2017) optimizer with a constant learning rate of $1 \times 10^{-4}$ and without weight decay. Models are trained with a batch size of $256$. Following SiT and FiTv2, we also apply an exponential moving average (EMA) with a decay factor of $0.999$ to the model weights during training, and we report the performance of the EMA checkpoints.

**Evaluation Metrics.** We use Fréchet Inception Distance (FID) (Heusel et al., 2017) as the primary evaluation metric. Inception Score (IS) (Salimans et al., 2016), sFID (Nash et al., 2021), improved Precision and Recall (Kynkäänniemi et al., 2019) are also included to holistically evaluate the generation quality. Moreover, the inference of DoD involves two processes – *sampling with class priors and sampling with visual priors*. Since the latter process is formally similar to image reconstruction, we also adopt reconstruction-FID (rFID), PSNR, and SSIM to assess the reconstruction quality, providing a more comprehensive evaluation of our method.

### 4.2 PRELIMINARY EXPERIMENTS

**Visual Prior Drastically Improves Performance.** We first verify how effectively visual priors can guide image generation.

We use the latents of ground truth images as inputs for the latent embedding module and report the reconstruction-FID (rFID) of the generated images. This experiment is conducted with a DoD-B model trained with $400K$ steps. As shown in Figure 4 (*Left*), classifier-free guidance (CFG) is critical for DoD, with a high CFG scale leading to significant improvements. The higher the CFG scale is, the more the model relies on the input condition, which contains the visual prior. When the CFG scale is set to $5.5$, the model performs best, achieving an rFID score of 3.6, which is much better than the score without CFG (10.8 rFID). This encouraging result demonstrates the potential of conditioning on visual priors. In this work, unless otherwise stated, we default to using a CFG scale of $5.5$ when employing image priors as conditions for DoD.

We also present the generation results of DoD models trained for $400K$ steps in Figure 4 (*Right*). The visual prior is significantly effective in improving the FID-50K score, since the performance of Stage 2 (*w/* visual prior) is obviously better than the performance of Stage 1 (*w/o* visual prior).

**Extracting Visual Priors Through Compression.** In Table 2, we evaluate the effectiveness of the compression-reconstruction approach used by DoD to extract visual prior information from three perspectives: generation, reconstruction, and linear probing. DoD controls the compression rate through the number of latent tokens $N$ in the latent embedding module, where using 256 tokens

Table 2: **Effects of the number of latent tokens $N$ in LEM.** We provide a comprehensive evaluation that includes: generation in Stage 1 (S1) and Stage 2 (S2), image reconstruction, and linear probing. The results are based on DoD-B, trained for $400K$ steps.

| $N$ | Generation | | | | Reconstruction | | | Linear Probe |
|---|---|---|---|---|---|---|---|---|
| | $\text{FID}_{S1}\downarrow$ | $\text{IS}_{S1}\uparrow$ | $\text{FID}_{S2}\downarrow$ | $\text{IS}_{S2}\uparrow$ | $\text{rFID}\downarrow$ | $\text{PSNR}\uparrow$ | $\text{SSIM}\uparrow$ | Top-1$\uparrow$ |
| 16 | 28.87 | 47.93 | 4.95 | **207.62** | 4.52 | 15.88 | 0.46 | **56.6** |
| 32 | **28.65** | **47.98** | **4.93** | 187.14 | 3.63 | 16.17 | 0.48 | 53.2 |
| 64 | 31.36 | 43.61 | 9.66 | 121.13 | 2.20 | 17.69 | 0.56 | 52.4 |
| 256 | 33.05 | 41.97 | 33.12 | 42.02 | **0.63** | **24.99** | **0.80** | 51.3 |

Table 3: **Effects of the joint training strategy in DoD.** We report the results based on DoD-B trained for $400K$ steps.

| $p_s$ | Generation | | | | Reconstruction | | |
|---|---|---|---|---|---|---|---|
| | $\text{FID}_{S1}\downarrow$ | $\text{IS}_{S1}\uparrow$ | $\text{FID}_{S2}\downarrow$ | $\text{IS}_{S2}\uparrow$ | $\text{rFID}\downarrow$ | $\text{PSNR}\uparrow$ | $\text{SSIM}\uparrow$ |
| 0.25 | 31.06 | 43.38 | 5.48 | 169.91 | **3.47** | **16.28** | **0.49** |
| 0.50 | 28.65 | 47.98 | 4.93 | 187.14 | 3.63 | 16.17 | 0.48 |
| 0.75 | **26.60** | **52.77** | **4.31** | **222.84** | 4.22 | 15.94 | 0.48 |

means no compression is applied. Thanks to the generative capabilities of the diffusion model, we can reconstruct images using only 16 latent tokens. We draw the following conclusions:

- A lower compression rate (*i.e.*, using more tokens) results in better reconstruction performance. This is intuitive, as a lower compression rate means that more information is retained for image restoration.

- More semantically rich representations can be learned with fewer tokens. Drawing from self-supervised learning, we measure the quality of the representations using linear probing accuracy. The best performance is achieved with 16 tokens, demonstrating that the latent embedding module in DoD effectively extracts global semantic information as visual priors through compression.

Interestingly, DoD with 32 tokens performs best in generation during both the first and second stages, suggesting a trade-off between reconstruction capability and semantic richness. By default, we utilize 32 latent tokens across all DoD variants.

**Joint Training in DoD.** In DoD, we implement a joint training strategy that optimizes both the backbone diffusion model and the latent embedding module concurrently. The training process is controlled by a hyperparameter $p_s$, which dictates the probability that the ground truth image priors are not utilized. The results of generation and reconstruction are detailed in Table 3. A higher value of $p_s$ leads to improved generation performance, while a lower value is more beneficial for reconstruction. As this work aims to validate the importance of visual priors rather than pursuing state-of-the-art performance, we simply set $p_s = 0.5$ as the default value.

Additionally, the joint training of DoD is based on the assumption that images of varying fidelity can provide similar visual priors through compression. We validate this assumption in Figure 4 (*Right*), where the output from stage 1 is used as a visual prior in stage 2, yielding more realistic images. However, stage 3, which relies on stage 2, does not show further improvements, indicating that the poor outputs from stage 1 and the good outputs from stage 2 provide similar prior information. Therefore, in our experiments, we report the performance of DoD from stage 2.

**Visual Prior Leads to Efficient Sampling in DoD.** One potential drawback of DoD is that it involves a multi-stage sampling process, which may result in a substantial computational overhead. Therefore, in Table 4, we demonstrate that *DoD outperforms the baseline model (i.e., FiTv2) even with fewer inference GFLOPs.* We employ the Euler sampler and fixed the number of sampling steps. The total GFLOPs is calculated as GFLOPs multiplied by the number of sampling steps. For methods utilizing classifier-free guidance (CFG), the total GFLOPs are doubled. For DoD, we

Table 4: **Comparison between DoD and FiTv2.** Both of the models are trained with 1.5M steps. We report the performance of DoD in Stage 1 (S1) and Stage 2 (S2) separately. With the same sampling steps and less computation, our DoD demonstrates better FID performance.

| Model | Steps | FID↓ | sFID↓ | IS↑ | Prec.↑ | Rec.↑ | Sampling Compute GFLOPs↓ |
|---|---|---|---|---|---|---|---|
| FiTv2-B | 60 | 19.67 | 6.45 | 73.34 | 0.61 | **0.66** | $1638_{(=27.3\times60)}$ |
| FiTv2-B-G (cfg=1.5) | 60 | 5.35 | **4.82** | 163.65 | 0.77 | 0.56 | $3276_{(=27.3\times2\times60)}$ |
| DoD-B (S1) | 30 | 23.43 | 7.36 | 65.85 | 0.59 | 0.65 | $795_{(=26.5\times30)}$ |
| DoD-B-G (S2, cfg=5.5) | $60_{=(30+30)}$ | **3.35** | 4.87 | **262.61** | **0.84** | 0.51 | $2409.6_{(=795+24.6+26.5\times2\times30)}$ |
| FiTv2-B | 240 | 18.66 | 6.15 | 73.96 | 0.61 | 0.66 | $6552_{(=27.3\times240)}$ |
| FiTv2-B-G (cfg=1.5) | 240 | 5.03 | **4.91** | 165.41 | 0.77 | 0.57 | $13104_{(=27.3\times2\times240)}$ |
| DoD-B (S1) | 120 | 20.81 | 6.39 | 67.67 | 0.60 | **0.67** | $3180_{(=26.5\times120)}$ |
| DoD-B-G (S2, cfg=5.5) | $240_{=(120+120)}$ | **3.13** | 5.44 | **254.34** | **0.82** | 0.53 | $9564.6_{(=3180+24.6+26.5\times2\times120)}$ |

Table 5: **Benchmarking class-conditional image generation on ImageNet** $256 \times 256$. "-G" indicates results with classifier-free guidance, "S2" denotes results from the second stage of DoD.

| Model | Images | Params | FID↓ | sFID↓ | IS↑ | Prec.↑ | Rec.↑ |
|---|---|---|---|---|---|---|---|
| BigGAN-deep | - | - | 6.95 | 7.36 | 171.40 | **0.87** | 0.28 |
| StyleGAN-XL | - | - | 2.30 | **4.02** | 265.12 | 0.78 | 0.53 |
| MaskGIT | 355M | - | 6.18 | - | 182.10 | 0.80 | 0.51 |
| CDM | - | - | 4.88 | - | 158.71 | - | - |
| ADM-G,U | 507M | 673M | 3.94 | 6.14 | 215.84 | 0.83 | 0.53 |
| LDM-4-G (cfg=1.5) | 214M | 395M | 3.60 | 5.12 | 247.67 | **0.87** | 0.48 |
| U-ViT-H-G (cfg=1.4) | 512M | 501M | 2.35 | 5.68 | 265.02 | 0.82 | 0.57 |
| Efficient-DiT-G (cfg=1.5) | - | 675M | 2.01 | 4.49 | 271.04 | 0.82 | 0.60 |
| Flag-DiT-G | 256M | 4.23B | 1.96 | 4.43 | **284.80** | 0.82 | 0.61 |
| DiT-XL-G (cfg=1.5) | 1792M | 675M | 2.27 | 4.60 | 278.24 | 0.83 | 0.57 |
| SiT-XL-G (cfg=1.5) | 1792M | 675M | 2.15 | 4.50 | 258.09 | 0.81 | 0.60 |
| FiT-XL-G (cfg=1.5) | 512M | 824M | 4.21 | 10.01 | 254.87 | 0.84 | 0.51 |
| FiTv2-XL-G (cfg=1.5) | 512M | 671M | 2.26 | 4.53 | 260.95 | 0.81 | 0.59 |
| FiTv2-3B-G (cfg=1.5) | 256M | 3B | 2.15 | 4.49 | 276.32 | 0.82 | 0.59 |
| DoD-S-G (S2, cfg=5.5) | 102.4M | 48M | 19.70 | 8.08 | 74.94 | 0.67 | 0.48 |
| | 256M | 48M | 11.97 | 6.23 | 108.95 | 0.74 | 0.48 |
| | 384M | 48M | 9.71 | 6.16 | 123.63 | 0.75 | 0.49 |
| DoD-B-G (S2, cfg=5.5) | 102.4M | 191M | 4.93 | 5.61 | 187.14 | 0.81 | 0.48 |
| | 256M | 191M | 3.42 | 5.65 | 236.59 | 0.82 | 0.52 |
| | 384M | 191M | 3.15 | 5.74 | 248.48 | 0.82 | 0.54 |
| DoD-XL-G (S2, cfg=3.5) | 102.4M | 613M | 2.34 | 5.00 | 228.42 | 0.78 | 0.61 |
| | 256M | 613M | **1.83** | 5.00 | 263.69 | 0.77 | **0.65** |

separately present the performance of two sampling stages, where the total GFLOPs for stage 2 are added to those of stage 1. Our observations are as follows:

- Increasing the sampling steps from 60 to 240 reduces the FID score across all cases.

- Due to the use of a shallower backbone, DoD at Stage 1 performs worse than FiTv2. However, in Stage 2, with the same number of sampling steps and fewer GFLOPs, DoD fully surpasses the baseline FiTv2. This further demonstrates the efficacy of the visual prior.

This experiment highlights the efficiency of DoD and the effectiveness of leveraging visual priors.

## 4.3 MAIN EXPERIMENTS

The comparison against state-of-the-art class-conditional generation methods is shown in Table 5. To ensure fair comparisons, we use the total number of training images (denoted as "Images" in the table) as a measure of the training cost. The total number of training images is calculated as training steps × batch size.

**DoD achieves better FID scores with lower training costs and much fewer parameters.** Specifically, using only $191M$ parameters, DoD-B achieves an FID score of 3.15, surpassing both ADM

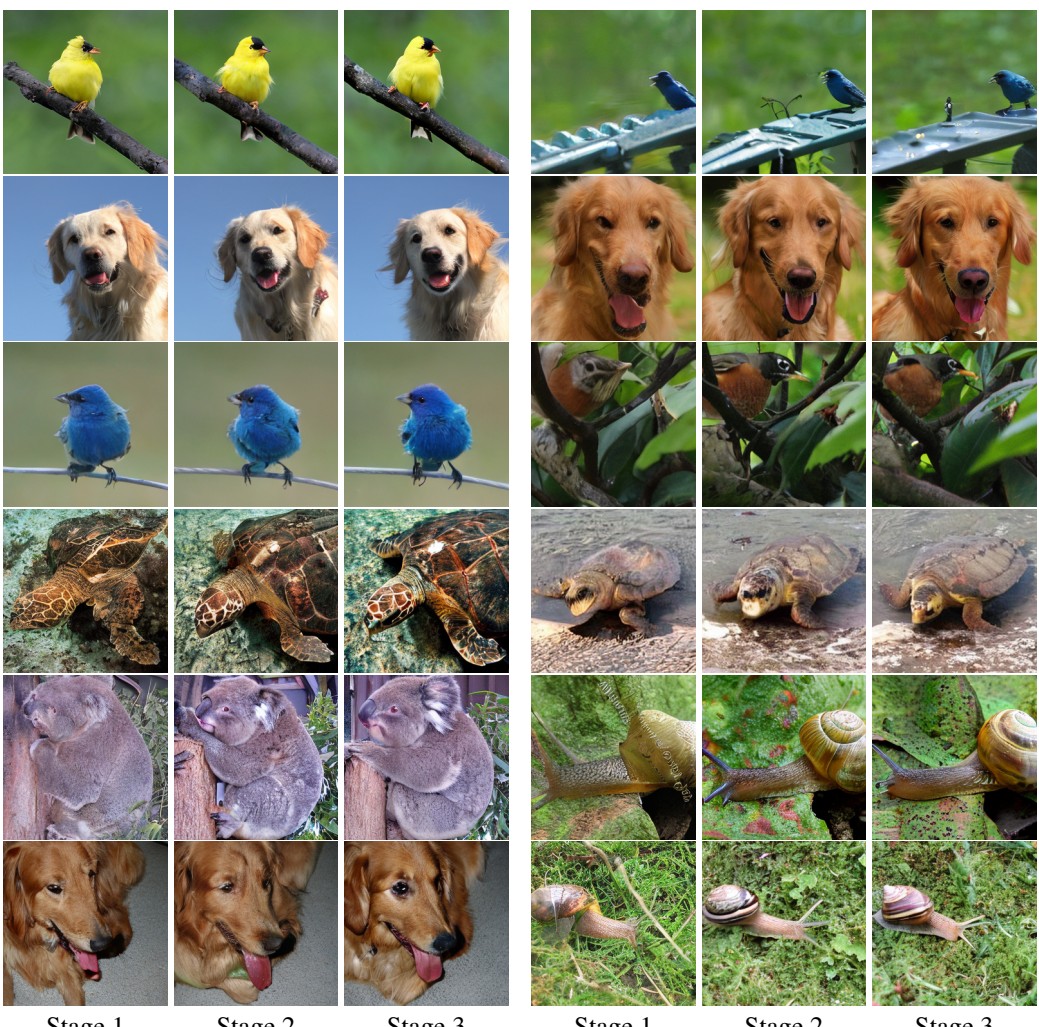

|         |         |         |         |         |         |
| Stage 1 | Stage 2 | Stage 3 | Stage 1 | Stage 2 | Stage 3 |

Figure 5: **Qualitative Results.** We present images generated by DoD-XL with $1M$ training steps. Across all stages, semantic information is preserved, and image quality improves progressively.

($3.94$ FID with $673M$ parameters) and LDM-4 ($3.60$ FID with $395M$ parameters). When scaling up the model, DoD-XL further improves performance to a FID of $1.83$, using only $613M$ parameters and $1M$ training steps, outperforming all previous diffusion models. Qualitative results are shown in Figure 1 and Figure 5.

## 5 CONCLUSION

In this work, we propose using visual priors to further enhance diffusion models. To achieve this, we introduce Diffusion on Diffusion (DoD), a novel multi-stage generation framework that provides rich guidance for the diffusion model by leveraging visual priors from previously generated samples. The latent embedding module in DoD effectively extracts semantic information from generated samples through a compression-reconstruction approach. DoD demonstrates remarkable training and sampling efficiency while being easy to reproduce. We conduct extensive experiments to validate the potential of injecting visual priors and explore the design space of the model. We hope our work will inspire the community to further investigate the use of visual priors in image generation.

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

# A APPENDIX

## A.1 TECHNOLOGIES ADOPTED BY FITV2

FiTv2 Wang et al. (2024) is an advanced diffusion transformer on class-guided image generation, evolving from SiT (Ma et al., 2024) and FiT (Lu et al., 2024a). The key modules of FiTv2 include 2-D Rotary Positional Embedding (2-D RoPE) (Su et al., 2024), Swish-Gated Linear Unit (SwiGLU) (Shazeer, 2020), Query-Key Vector Normalization (QK-Norm), and Adaptive Layer Normalization with Low-Rank Adaptation (AdaLN-LoRA) (Hu et al., 2022).

**2-D RoPE.** FiTv2 follows FiT and adopts 2-D RoPE as its positional embedding. RoPE applys a rotary transformation to the embedding, incorporating both absolute and relative positional information into the query and key vectors. Benefiting from such property, RoPE and its high-dimensional variants have been widely adopted in current vision diffusion transformer models (Lu et al., 2024a; Gao et al., 2024; Esser et al., 2024b).

**SwiGLU.** SwiGLU is widely used in advanced language models like LLaMA (Touvron et al., 2023a;b; Dubey et al., 2024). FiTv2 utilizes SwiGLU module as its Feed-forward Neural Network (FFN), rather than normal Multi-layer Perception (MLP). The SwiGLU is defined as:

$$\begin{aligned} \text{SwiGLU}(x, W, V) &= \text{SiLU}(xW) \otimes (xV) \\ \text{FFN}(x) &= \text{SwiGLU}(x, W_1, W_2)W_3 \end{aligned} \tag{8}$$

**QK-Norm.** FiTv2 applies LayerNorm (LN) to the Query (Q) and Key (K) vectors before the attention calculation. This technique effectively stabilizes the training process, particularly during the mixed-precision setting, as well as slightly improves the performance. Formally, the attention is calculated as:

$$\text{Softmax}(\frac{1}{d_k}\text{LN}(Q_i)LN(K_i)^T). \tag{9}$$

**AdaLN-LoRA.** FiTv2 utilizes AdaLN-LoRa to reduce the too many parameters occupied by the original AdaLN module in each transformer block. Besides, a global AdaLN module is employed to extract overlapping condition information and reduce the information redundancy in each block. Let $S^i = [\beta_1^i, \beta_2^i, \gamma_1^i, \gamma_2^i, \alpha_1^i, \alpha_2^i] \in \mathbb{R}^{6 \times d}$ denote the tuple of all output scale and shift parameters, $\mathbf{c} \in \mathbb{R}^d$ and $\mathbf{t} \in \mathbb{R}^d$ represent the embedding for class and time step respectively. In FiTv2 blocks, $S^i$ is calculated as:

$$\begin{aligned} S^i &= \text{AdaLN}_{\text{global}}(\mathbf{c} + \mathbf{t}) + \text{AdaLN}_{\text{LoRA}}(\mathbf{c} + \mathbf{t}) \\ &= W^{\text{g}}(\mathbf{c} + \mathbf{t}) + W_2^i W_1^i(\mathbf{c} + \mathbf{t}), \end{aligned} \tag{10}$$

where $W^{\text{g}} \in \mathbb{R}^{(6 \times d) \times d}, W_2^i \in \mathbb{R}^{(6 \times d) \times r}, W_1^i \in \mathbb{R}^{r \times d}$, and the bias parameters are omitted for simplicity.

**Logit-Normal Sampling.** Despite these advanced modules, FiTv2 adopts the Logit-Normal sampling (Esser et al., 2024b) strategy to accelerate the model convergence. Normally, rectified flow models sample times uniformly from the $[0, 1]$ interval, which means each part of the noise scheduler is trained equally. FiTv2 samples timesepts from a logit-normal distribution (Atchison & Shen, 1980), which is formulated as:

$$u \sim \mathcal{N}(\mathbf{0}, \mathbf{1}), \quad t = \log(\frac{u}{1 - u}) \tag{11}$$

where $\mathcal{N}(\mathbf{0}, \mathbf{1})$ denotes the standard normal distribution. This sampling strategy puts more attention on the middle part of the sampling process, as recent studies (Karras et al., 2022; Chen, 2023) have disclosed that the intermediate part is the most challenging part in diffusion process.

## A.2 MORE QUALITATIVE RESULTS

We present additional qualitative results generated by DoD-B and DoD-XL, displayed in Figure 6 and Figure 7, respectively.

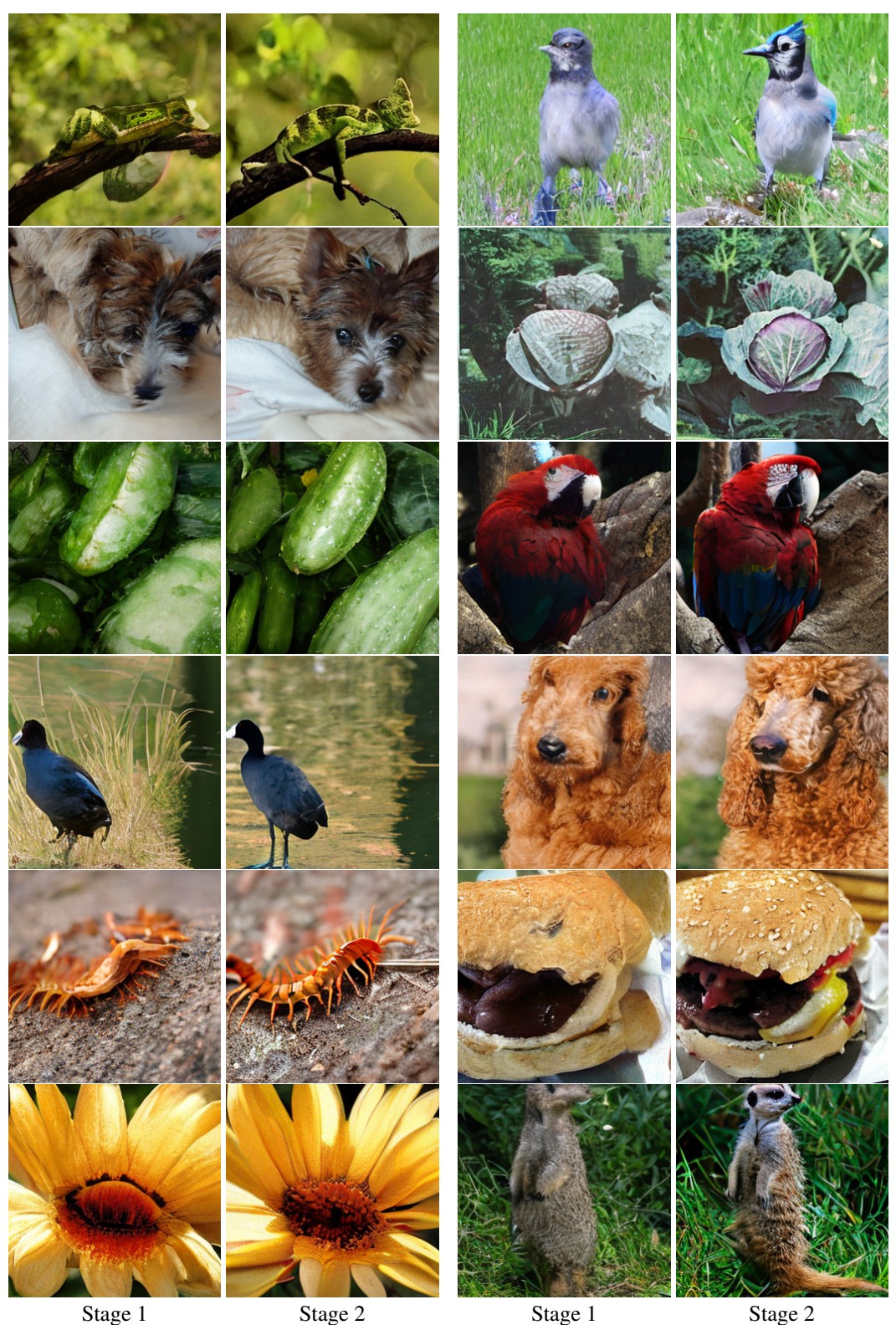

Stage 1     Stage 2     Stage 1     Stage 2

Figure 6: **Images generated by DoD-B.**

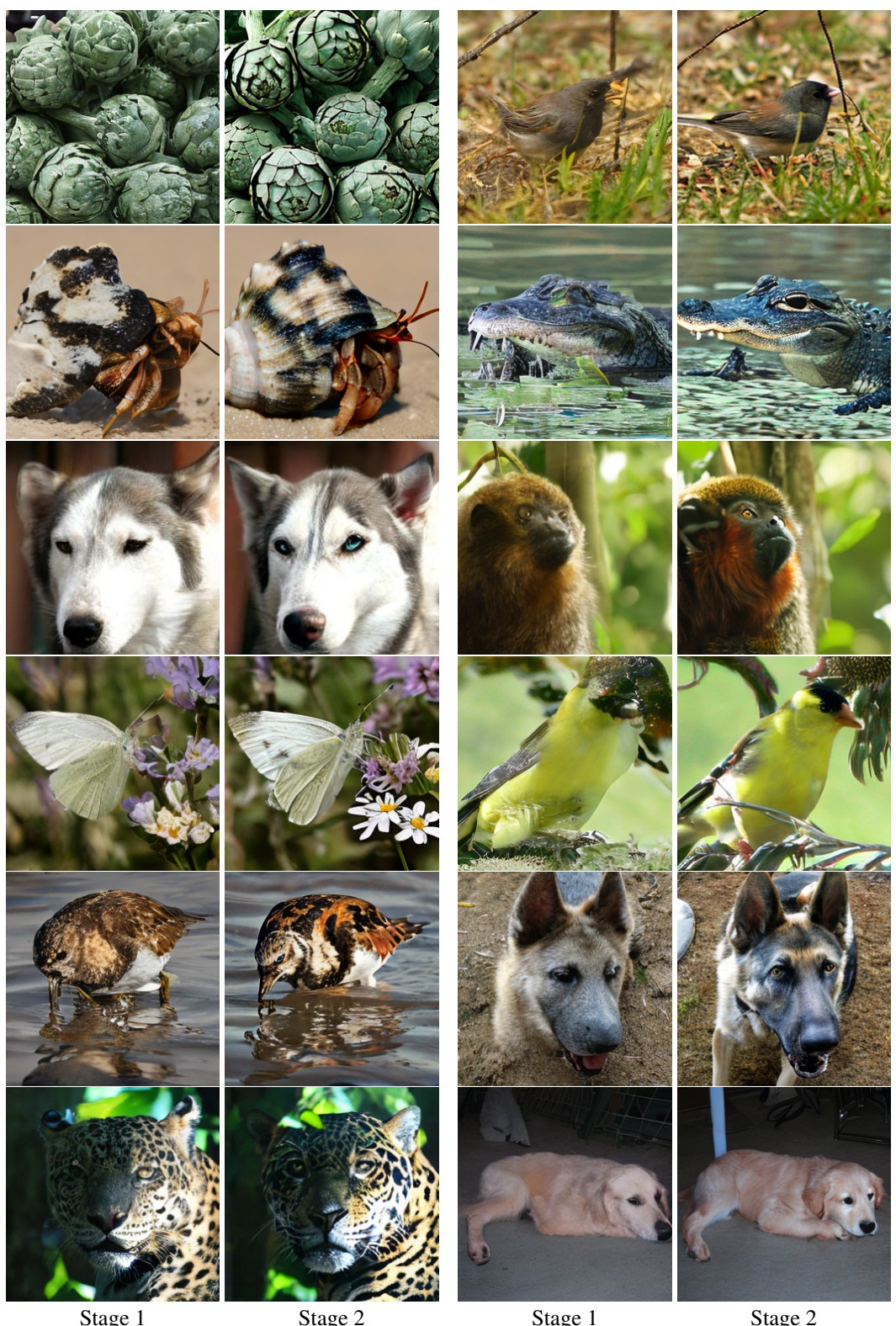

| Stage 1 | Stage 2 | Stage 1 | Stage 2 |

Figure 7: **Images generated by DoD-XL.**

