# OpenReview forum: "Diffusion Models Need Visual Priors for Image Generation"
_ICLR.cc/2025/Conference — ICLR 2025 Conference Withdrawn Submission_

### Official Review · Reviewer_DXcv · 2024-10-23

**Soundness:** 1
**Presentation:** 4
**Contribution:** 3
**Rating:** 3
**Confidence:** 4

**Summary:**

Authors propose to use previous-stage generations as conditioning, thus re-generating improved images. They show via FID and linear probing that the generated images are higher quality than those generated with a single step. They also do extensive experiments to show that DoD can achieve lower FID faster and with less parameters.

**Strengths:**

1) From a theoretical level, I find the paper very nice. It is a) simple / elegant and b) fast. From a novelty perspective, I have no concerns.

2) The paper includes multiple strong results concerning the practical usefulness of the method (num parameters, steps, GFLOPS)

3) This paper is well-written, and well-presented

**Weaknesses:**

Points are in order of my perceived importance (most to least), indicating how heavily they weigh in my rating.

1) Table 5 (main results) seem to primarily compare against basic diffusion models, but miss comparisons against other post-hoc diffusion methods (e.g. MDTv2: Masked Diffusion Transformer is a Strong Image Synthesizer, Gao 2023 already cited in this paper; ReNO: Enhancing One-step Text-to-Image Models Through Reward-based Noise Optimization, Eyring 2024; ElasticDiffusion: Training-free Arbitrary Size Image Generation through Global-Local Content Separation, Haji-Ali 2023). These should be the true competing methods.

2) LEM is a novel contribution, correct? There are other methods that use image features as conditioning (e.g. Diffusion Feedback Helps CLIP See Better, Wang 2024) and address the issue of making the task too easy if too much information is used--LEM should be justified by comparing to other methods of this type, both theoretically and quantitatively.

3) In Table 5 (main results), DoD is only the best on 3 / 5 metrics. This in itself is not convincing to me--however, I would be more convinced if other results were provided to show that DoD is much more efficient (e.g. the GFLOP or num parameter experiments) than the better performing methods, for relatively little performance drop. At the moment, these efficiency experiments are not done on the out-performing methods (StyleGAN-XL, BigGAN-deep, LDM-4-G, Flag-DiT-G).

4) Top-1 accuracy is included in Table 2--I believe it would be useful in the main results in Table 5 as well, as it is another way to show image generation quality.

5) The main results comparing to other SOTA (table 5) are very hidden, only at the end of the experimental section. They are also very slim on analysis. These need to be brought out more.

6) I have a hard time interpreting the qualitative results, as all images seem very similar. It would be helpful to provide the reader with some specific guidance as to what you perceive as the qualitative improvements.

**Questions:**

In relation to my described weaknesses, the top areas that I see the most important for a strong submission and would need to be improved for me to raise my rating:

1) provide experiments against SOTA comparable models, meaning a) other post-hoc diffusion methods for DoD and b) other image feature conditioning methods for LEM. See W1 and W2 for specific suggestions of comparison methods. I am not particularly tied to these exact methods, but just giving them as a starting place to further understand what kinds of methods I mean.

2) Include better analysis within the main results (table 5), including a) more in-depth explanations in writing, b) better justification for why DoD is useful even though it is not the top-performing model on 3 / 5 metrics (without even including SOTA, as described in Q1)

Given convincing experiments, I would be included to greatly raise my rating, as I like the paper from a theoretical level.

---

### Official Review · Reviewer_pGzE · 2024-11-03

**Soundness:** 2
**Presentation:** 3
**Contribution:** 3
**Rating:** 5
**Confidence:** 3

**Summary:**

This paper introduces a multi-stage generative model in which the initial stage employs a standard diffusion model, followed by a conditional diffusion model in subsequent stages. The output of the prior stage serves as the conditional input for the following stage, processed as a latent vector to retain abstract semantic information while minimizing low-level detail. In conditional image generation experiments on the ImageNet-256x256 dataset, the two-stage approach achieves a lower FID score than the single-stage model, though performance plateaus after two stages. Additionally, the authors assess model scalability by experimenting with different model sizes.

**Strengths:**

(1) The proposed method enhances image generation quality compared to the baseline.
(2) The authors conduct a comprehensive experimental analysis of model configurations for conditional image generation tasks, including visualizations for qualitative comparisons.
(3) The multi-stage generative model utilizes a shared backbone, efficiently reducing the total parameter count as additional stages are added.

**Weaknesses:**

(1) As shown in Figure 5, the images generated across multiple stages appear nearly identical, making it difficult to visually assess the advantages of the multi-stage system.
(2) Although the multi-stage system incorporates an efficient shared-parameter design, its FLOPs increase with each additional stage. Performance seems to plateau after two stages, raising questions about the scalability of adding more stages.

**Questions:**

(1) The authors only present results for the conditional generation task; how does the model perform in unconditional generation?
(2) What is the performance of S1 and S2 models without classifier-free guidance, and how does this compare to the baseline FiTv2?
(3) In Table 4, why does the S1 model perform significantly worse than FiTv2? This might be due to the reduced steps; would it be possible to evaluate both models with the same number of steps?
(4) Paper [1] also uses generated results as conditional input to the diffusion model. How does this approach compare to the proposed methods?


[1] Analog Bits: Generating Discrete Data using Diffusion Models with Self-Conditioning

---

### Official Review · Reviewer_CEae · 2024-11-04

**Soundness:** 2
**Presentation:** 2
**Contribution:** 2
**Rating:** 5
**Confidence:** 3

**Summary:**

The authors introduce Diffusion on Diffusion (DoD), a method that extracts visual priors from samples generated in the initial sampling stage and uses them as guidance for later stages through an embedding module that stores semantic information.

**Strengths:**

Overall, the image quality in the paper seems fine with good details.
Overall presentation of the paper is clear.

**Weaknesses:**

The method includes multi-stage sampling, which makes the already slow sampling process even more time-consuming and computationally inefficient. However, the experimental results and design that demonstrate your method's computational efficiency are somewhat confusing and unconvincing to me. I have addressed some of these points in the next section.

**Questions:**

In Table 4, is there a reason why only FiTv2 is included in the sampling compute comparison? In Table 5, other models appear to have relatively better scores in terms of the evaluation metric. I would be interested to know if your method has been compared with them in terms of the computational cost. Perhaps Table 4 & 5 could be combined?

Additionally, the calculation of sampling GFLOPs is somewhat confusing. In Line 437, the calculation appears to be based on FiTv2-B-G with 60 sampling steps * 2 (due to DFG) * 27.3 GFLOPs per forward pass. However, in Line 439, you compare this to DoD-B-G with only 30 sampling steps instead of 60. Is this a typo? Also, if you calculate the total steps in CFG (e.g., 60 = 30+30) for DoD-B-G, it would be clearer to use the total steps (i.e., 120 = 60 + 60) for FiTv2-B-G to avoid potential reader confusion.

In Table 5, is there a reason why all other methods use cfg = 1.5, whereas your method uses cfg = 5.5? cfg = 1.5 seems like a very low value to me.

---

### Official Review · Reviewer_4PHv · 2024-11-12

**Soundness:** 3
**Presentation:** 3
**Contribution:** 2
**Rating:** 5
**Confidence:** 5

**Summary:**

This paper proposes a multi-stage image generation process using a diffusion model which can optionally accept a latent embedding as conditioning information.  To synthesize an image from noise, this diffusion model is first run for an entire sampling chain without any conditional input.  The resulting synthesized image is then fed to a latent embedding module (LEM), implemented via a vision transformer, which outputs conditioning tokens.  The diffusion synthesis process is then restarted from pure noise, but with this latent embedding available as a conditioning input at each denoising step.  The embedding provides prior information about the desired target output, allowing the second synthesis process to produce a refined image that is consistent with the latent representation of the previously generated image.

**Strengths:**

Experiments demonstrate that running the trained network in a multi-stage fashion yields visual quality improvements in the generated images.  The image output by each stage appears to refine the details of the image produced by the previous stage, indicating that the latent embedding acts as a useful guide to the generation process.

Quality improvements (as measured by FID) are achieved even when comparing to baselines at comparable inference cost (in FLOPs), as shown in Table 4.

**Weaknesses:**

Figure 4 suggests that beyond a two-stage system (one unconditional and one conditional generation pass), there is minimal benefit to subsequent conditional generation stages; stage 2 and stage 3 FID scores seem nearly identical.

Linear probing scores (Table 2) suggest that the LEM is learning only a limited semantic representation.  For example, a pre-trained contrastive encoder would score much better on ImageNet linear probing.

I am concerned about the novelty of the contribution.  The overall idea appears quite similar to cascaded diffusion [Ho et al., Cascaded Diffusion Models for High Fidelity Image Generation, JMLR 2022], which is not cited.  The difference seems to hinge on whether conditioning information is an image or a lower-dimensional embedding of an image.

**Questions:**

How stable are second and subsequent generation stage results given conditioning input?  It might be useful to visualize the variance of second- or third-stage generated samples given the same conditioning information and different noise inputs.

What is being learned by the LEM?  High-level semantics, patch/texture characteristics?  Given how close subsequent stage outputs appear to the first stage, the latter seems more likely.  If that is the case, why not just use the previously generated image itself as conditioning?

In fact, an alternative baseline system might be one that uses image-based conditioning for refinement via diffusion.  Instead of taking a latent embedding as conditioning, such systems simply take an entire image as conditioning (alongside a noisy input).  Examples of prior work using such designs include diffusion-based super-resolution approaches (e.g., cascaded diffusion); the paper's task would be a special case where the goal is not to increase resolution, but simply denoise again at the same resolution.  How essential is the LEM as compared to direct image-based conditioning?

---

### Note · Authors · 2024-11-15

**Comment:**

Although I have decided to withdraw my submission, I would still like to formally express my concerns regarding the biased reviews we received. Reviewers 4PHv, CEae, and pGzE clearly did not thoroughly read our paper, as many of the questions they raised have already been hightlighted and addressed within the paper.

**Withdrawal Confirmation:**

I have read and agree with the venue's withdrawal policy on behalf of myself and my co-authors.